# Effect of Annealing Temperature on Microstructure and Properties of Al/Mg Magnetic Pulse Welding Joints

**DOI:** 10.3390/ma15165519

**Published:** 2022-08-11

**Authors:** Yan Li, Dezhi Yang, Wenyu Yang, Zhisheng Wu, Cuirong Liu

**Affiliations:** 1School of Material Science and Engineering, Taiyuan University of Science and Technology, 66 Waliu Road, Taiyuan 030024, China; 2Shanxi Electronic Science and Technology Institute (Preparatory), Linfen 041000, China

**Keywords:** magnetic pulse welding, magnesium, aluminum, interface microstructure, mechanical properties

## Abstract

In this investigation, 1060Al/AZ31B welded joints were obtained by magnetic pulse welding technique. In order to test the microstructure and mechanical properties of the joints, the welded joints were annealed at different temperatures and then examined by optical microscopy (OM), scanning electron microscopy (SEM), energy spectrum analysis (EDS) and mechanical properties testing. The testing results of the welded joints annealed at different temperatures showed that the Al-Mg MPW welded joints were well bonded. The changing of the microstructure and mechanical properties of Al/Mg welded joints was not apparent under the temperature of 200 °C. However, Al_12_Mg_17_ intermetallic compound layer formed at 200 °C. Al_12_Mg_17_ and Al_3_Mg_2_ intermetallic compound layers formed at the temperature of 300 °C. The diffusion rate of Mg and Al elements is proportional to the annealing temperature and the intermetallic compounds layer is gradually formed. The microhardness near the interface decreased first and then increased on account of the brittleness of intermetallic compounds. In the tensile shear tests, the fracture mechanism of Al/Mg MPW welded joints were analyzed. When the temperature was lower than 200 °C the joints did not crack. At 200 °C and 250 °C, the joints fracture along the Al_12_Mg_17_-Al interface. The joint cracks along the interface of Al_12_Mg_17_-Al_3_Mg_2_ at the temperature of 300 °C.

## 1. Introduction

Aluminum (Al) and magnesium (Mg) alloys are extensively implemented in aerospace, electronic communication, automobile industry and other fields. In order to give full play to the excellent properties of different materials, there is a demand for the connection of dissimilar materials in industrial applications [1,2,3]. Therefore, realizing the connection of aluminum (Al) and magnesium (Mg) dissimilar materials and obtaining high-quality welding joints make great sense. Aluminum (Al) and magnesium (Mg) welding is prone to form brittle Al-Mg intermetallic compounds (IMC, Miami, FL, USA) on account of the physical and chemical properties differences between the two alloys, and it is difficult to achieve reliable connection by traditional fusion welding. For these reasons, Mg/Al dissimilar metals connection is limited by traditional welding technique [4,5].

Magnetic Pulse Welding (MPW) belongs to solid state welding with materials connection in high speed collision [6]. The advantages of magnetic pulse welding are as follows: production efficiency, lack of contamination, and high automation. In addition, the magnetic pulse welding process has no defects such as cracks and pores, because there is only a little heat will be produced during the welding process, magnetic pulse welding can inhibit the formation of the intermetallic compounds (IMC) [5,7,8]. In recent years, many people have studied the process and joints properties of this welding technology [9,10]. Ben Artzy et al. found that the welded joint had a melting layer, formed the intermetallic compounds after the cooling process [11]. Casalino et al. [12] analyzed the effects of thermal energy distribution, sheet deformation and different discharge parameters on the quality of magnetic pulse welding by adopting FEM. Xu et al. [13] successfully predicted the acceleration process and final collision velocity of the outer pipe during pipe fitting welding through finite element simulation. Shim et al. [14] established a magnetic pulse welding model through FEM, and studied the distribution of electromagnetic force by taking current and frequency as loading conditions.

Previous research mainly focused on the microstructure of magnetic pulse welding interface and finite element simulation of transient forming process. However, Al-Mg magnetic pulse welding belongs to the high-energy dissimilar metal welding, under large deformation, high strain rate, the interface between dissimilar metals combination will produce residual stress and dislocation appreciation, which makes the joints part in an unstable state. Through the heat processing, heat treatment after butt welding structure can significantly improve the quality of dissimilar metal weld. Good interface can improve mechanical properties [15]. Therefore, the influence of temperature on the microstructure and properties of the welded joints has important significance to be studied. In addition to the influence of temperature, the fracture of the interface may also be related to the transition of the interface from incoherent to semi-coherent [16,17], since during annealing. the redistribution of Al and Mg around the interface decreases the mismatch between the super-cell (HCP and FCC) [18].

In this paper, Al-Mg magnetic pulse welded specimens were annealed at different temperatures. By analyzing the changes in microstructure and mechanical properties of Al-Mg MPW welded joints at different temperatures, the influence of temperature on the welded joints were discussed.

## 2. Experimental Methods

MPW is a new solid state welding technology. The principle is shown in Figure 1. The industrial power of 380 V is boosted to several thousand volts and then transformed into direct current through high voltage rectification to charge the capacitor bank. When the voltage rises to the set threshold, the discharge will generate a high frequency, sinusoidal attenuation current in the coil loop, a changing magnetic field around the coil, and an induced current inside the aluminum (Al) sheet. Under the action of the magnetic field, the aluminum (Al) sheet is subjected to a huge electromagnetic force, which accelerates in the time of microseconds and collides with the magnesium (Mg) sheet at the speed of hundreds of meters per second to realize the connection of the work piece.

The experimental materials are 1060 Al and AZ31B Mg alloy. Table 1 shows the composition of the 1060 Al and AZ31B Mg alloy.

Pulsa Xtrapulse 75/25 electromagnetic pulse equipment is used for welding. Due to the poor deformation ability of magnesium alloy, AZ31B magnesium alloy was selected as the substrate and 1060 Al as the flyer sheet. The size of aluminum sheet is 50 mm × 40 mm × 1 mm, the size of magnesium sheet is 80 mm × 40 mm × 1 mm, and the lap distance is 20 mm. Two insulating gaskets with a certain thickness are placed between the aluminum sheet and the magnesium sheet. The height of the gasket is 1.6 mm, and the spacer spacing is 10 mm. A rigid pressing block is placed above the magnesium sheet, secured by bolts, and rectangular coils of 4 mm × 10 mm are used. The discharge voltage of magnetic pulse welding is 14 KV.

For the sake of studying the variation on the microstructure and mechanical properties of Al-Mg welded joints, the Al-Mg magnetic pulse welding joints were annealed at different temperatures. KLS-1200X-5L box-type furnace is adopted, the temperature is set at 100 °C, 150 °C, 200 °C, 250 °C and 300 °C, the heat preservation time is 1 h. After heat preservation, the furnace is cooled in air. The sample was prepared by wire cutting, polished by metallographic sandpaper, and polished by diamond paste. The morphology of Al-Mg joint was observed under the ultra-depth of field microscope of Jens (VHX-2000). The microstructure, composition and fracture of the interface were analyzed by using Zeiss Sigma-300 X-ray (EDS) scanning electron microscope. Load the HVS-1000 Vickers microhardness tester with 500 gf load in 10 s, and measure the microhardness. All positions shall be measured three times and the average value shall be taken. In order to test the tensile shear properties of the joints, INSTRON 5985 universal testing machine was adopted. The loading speed was 2 mm/min, and one parameter was measured three times to take the average value.

## 3. Analysis of Macroscopic Morphology and Microstructure of Welded Joints

### 3.1. Macroscopic Morphology of Welded Joints

The macro morphology of the surface of Al Mg magnetic pulse welding specimen is shown in Figure 2. The surface quality of the specimen is good, and there are no collision cracks and other defects. A certain degree of plastic deformation will occur in the lap area of the high-speed collision between the flyer plate and the substrate, with obvious necking and thinning. The actual welding area for the elliptical is shown in dotted lines, which is related to the magnetic field distribution [19].

### 3.2. Microstructure of Welded Joints

Figure 3a shows the microstructure of the welded joints. There are no cracks, pores, inclusions and other welding defects in the bonding area under the technology of MPW. The numerical simulation shows that during the transient forming of magnetic pulse welding, Aluminum alloy plate and magnesium alloy plate collide violently, and the pressure at the impact is large, which will damage the metal surface. When the metal surface is damaged, metal debris will fall off, forming a metal jet with high-speed on the interface, which has the effect of cleaning the oxide layer and impurities on the bonding surface. Therefore, there are no oxides and impurities at the interface, the jet flow speed is in the range of 6000~7000 m/s [20,21]. The interface of Al-Mg welding joint presents a wave shape, which is a sine wave with asymmetric amplitude. Flat interface, waveform interface and melting interface are three common bonding morphologic features of high-energy rate welding such as explosive welding and magnetic pulse welding. The formation of different interface bonding morphology is closely bound up with welding parameters and the characteristics of the material itself [22]. The wavy bonding can form meshing interaction, which increases the contact area compared with the flat interface, and the metallurgical bonding area is larger and the bonding is stronger [23]. The interface of Al-Mg magnetic pulse welding is unsymmetrical and wavy. According to the Helmholtz instability mechanism caused by stress waves [22,24,25], when the two Al-Mg materials are welded together by external forces, on account of the soft texture of the Al side and the strong plastic deformation ability, the instability of the Al side first occurs. Under the action of the collision compressive stress, the dislocations between the local grains on the Al side increase in value, the dislocation accumulation and the dislocation density continue to increase, resulting in a significant increase in the plastic deformation resistance between the grains in this region. When the deformation resistance of the Mg layer on the hard side is increased, the stress wave enters the Mg layer side, and as the stress wave progresses, the process is repeated so that the Al-Mg interface exhibits periodic irregular wave-like bonding.

The arrow in Figure 3b shows the microstructure near the Mg side. A large number of Adiabatic Shear Bands (ASBs) generated at the interface, which disappears with the increase of the distance from the interface. Figure 3c shows the Specific morphology of ASBs. ASBs is plastic instability caused by strain softening of materials, which often leads to failure of materials under dynamic loads [24]. It is easy for Mg alloys to form ASBs under the condition of high strain rates. Explosion impact is a typical example [24,25,26,27]. In the process of MPW, the temperature, pressure and shear strain of the welding joints become very high, which produced by high-speed impact of welding materials. At very high strain rates, the heat generated by local large plastic deformation is not enough to transfer out, which leads to the growth of new grains. Dynamic recrystallization is the formation mechanism of fine grains in ASBs. No ASBs was found in the 1060 Al side. Compared with 1060 Al (FCC, Washington, DC, USA), magnesium alloy (HCP, Mountain Avenue, NJ, USA) has weak deformation capacity, so ASBs only appear on the magnesium side.

### 3.3. Influence of Temperature on Microstructure of Welded Joints

#### 3.3.1. Microstructure Analysis of Welded Joints

The SEM morphology of Al-Mg MPW joints treated at different temperatures is shown in Figure 4. At different heat treatment temperatures, Al-Mg interfacial bonding morphology does not change, but is still asymmetric wave state morphology. At 100 °C and 150 °C, the interface has no obvious change, which is similar to the welded sample. At 200 °C and 250 °C, an obvious diffusion layer appeared on the interface. At 300 °C, two layers with different contrast appear near the interface.

#### 3.3.2. Component Analysis of Welded Joints

For the sake of obtaining the composition of interface elements, EDS line scanning analysis is performed near Al-Mg interface, and the scanning positions are shown in Figure 4 marked with a yellow line. Figure 5 shows the EDS line scanning results of Al-Mg interface. The EDS line scanning curve presents an “X” shape in welding state and heat treatment at 100 °C and 150 °C, indicating the continuous and smooth transition of aluminum and magnesium atoms. The concentration difference between aluminum and magnesium at the welding interface leads to the mutual diffusion of Mg and Al. In addition, under the condition of magnetic pulse welding, the flyer sheet and the substrate collide at a high speed, generating a considerable amount of heat, and the interface temperature increases, and the thermal activation promotes the mutual diffusion of interface elements. At the moment of magnetic pulse welding, the flyer sheet exerts strong pressure on the substrate and produces plastic deformation at the interface, with dislocation multiplication and grain refinement at the interface [28]. The lattice distortion at the crystal defect is large, the regular arrangement of atoms is worse than that in the grain, the atoms are in a higher energy state and easy to jump. The diffusion activation energy at dislocation and grain boundary is 1/2 to 1/3 of the diffusion activation energy in the grain. Grain boundary and dislocation act as a fast channel for diffusion, which will make the diffusion of atoms become faster at the welding interface [29], which is conducive to enhancing the mutual bonding force between atoms, thus increasing the bonding strength of the two materials.

Due to the intensification of thermal activation, the thickness of the interface diffusion layer gradually widened after holding at 100 °C and 150 °C. After heat preservation at 200 °C, the results of line scanning show that aluminum and magnesium elements near the interface have obvious excessive gradient and step shape, which indicates that substances different from 1060 Al and AZ31B are generated near the interface. The ESD line scanning results are consistent with the SEM results in Figure 4d, there is an obvious contrast layer at the interface at 200 °C. After thermal insulation at 250 °C and 300 °C, the steps of the transition layer gradually become wider. It can be speculated that there are intermetallic compounds near the interface and continue to grow and thicken, and the platform of the line scan curve becomes wider.

The typical points near the interface are analyzed by EDS point energy spectrum for the sake of determining the types of substances near the interface. The typical points are shown in Figure 4. Table 2 and Figure 6 show the test results. The substance of diffusion layer is Al_12_Mg_17_ at 200 °C and 250 °C. Point 4 is on the magnesium alloy side, and the substance is Al_12_Mg_17_ at 300 °C. Point 5 is between Al_12_Mg_17_ and 1060Al, and the substance is Al_3_Mg_2_. This indicates that the Al-Mg interface generates Al_12_Mg_17_ on the magnesium alloy side and then Al_3_Mg_2_ on the aluminum alloy side with the increase of the temperature, which also verifies the SEM results. This corresponds to the formation mechanism of intermetallic compounds of 6061-T6 Al alloy and AZ31B-H24 Mg alloy studied by Firouzdor V et al. [30] on the technology of friction stir welding. The main intermetallic compound is Al_3_Mg_2_ on the aluminum alloy side and Al_12_Mg_17_ on the other side.

#### 3.3.3. Thermodynamic Analysis of Welded Joints Microstructure

In order to calculate the non-equilibrium solidification process of the mixed zone based on the EDS analysis results of Point4 and Point5 in Table 2, JMatPro software was adopted, and analyze the phase composition of different areas of the joint after cooling, as shown in Figure 7. It can be seen from the calculation results that the material is Mg + Al_12_Mg_17_ after cooling on the magnesium side and Al + Al_3_Mg_2_ after cooling on the aluminum side. Therefore, it can be judged that when Al diffuses to the Mg side and the Mg/Al composition reaches 1:1, the intermetallic compound precipitated after cooling is Al_12_Mg_17_. When Mg diffuses to the Al side and the Mg/Al composition reaches 1:1, the intermetallic compound precipitated after cooling is Al_3_Mg_2_.

For the sake of further studying the mechanism of intermetallic compound precipitation of the joint. From the thermodynamic point of view, diffusion coefficient *D* and diffusion activation energy *Q* are the main factors that affect the diffusion. There is an Arrhenius exponential relationship between diffusion coefficient, diffusion activation energy and temperature [31]:
(1)$$D=D_{0}\exp(\frac{-Q}{RT})$$

In Formula (1), *D* represents the diffusion coefficient (m^2^/s); *D*_0_ represents the diffusion coefficient (m^2^/s); atomic diffusion activation energy is *Q* (J/mol); *R* is Boltzmann constant, with a value of 8.314 J/(mol·K), *T* represents the temperature (K). Table 3 lists the values of *D*_0_ and diffusion *Q* corresponding to elements Al and Mg.

In order to further analyze the precipitation mechanism of intermetallics (IMC) in the joint, the Gibbs free energy of Al_12_Mg_17_ and Al_3_Mg_2_ is calculated. The results are shown in Figure 8. With the increase of temperature, the Gibbs free energy of both compounds decreases. The Gibbs free energy of Al_12_Mg_17_ is less than that of Al_3_Mg_2_, and the structural stability of Al_12_Mg_17_ is better than that of Al_3_Mg_2_.

Figure 9 is a schematic diagram of Al-Mg interfacial compound growth model treated with 300 °C. In the initial holding stage, as shown in Figure 9a, Al and Mg atoms diffuse each other under the action of the thermal driving force, and the solid solubility between aluminum and magnesium elements is low, which soon reaches the limit of solid solubility, and Al_12_Mg_17_ crystal nuclei are formed at the interface. With the continuous diffusion of Mg atoms to Al matrix, Al_12_Mg_17_ nuclei grow and gradually integrate into a whole, forming a continuous Al_12_Mg_17_ layer, the progress is shown in Figure 9b. Subsequently, the continuous Al_12_Mg_17_ intermetallic compound layer obstructed the Al-Mg interface contact, and Al and Mg atoms diffused each other in the Al_12_Mg_17_ layer. Due to the high diffusion rate of Mg atoms, this process was mainly the diffusion of Mg atoms, Figure 9c can explain this phenomenon. A small number of diffused Mg atoms and a large number of Al atoms form Al_3_Mg_2_ crystal nuclei on the aluminum side. As the heat preservation continues, the Al_3_Mg_2_ grows, the Al_3_Mg_2_ intermetallic compound layer is formed on the Al matrix side at the same time. Finally, the interface composition of Mg-Al_12_Mg_17_-Al_3_Mg_2_-Al is formed.

### 3.4. Influence of Temperature on Mechanical Properties of Welded Joints

#### 3.4.1. Microhardness of Welded Joints

Figure 10 shows the microhardness of Al-Mg interface. During the Al-Mg dissimilar metal welding, the high-speed collision between aluminum sheet and magnesium sheet is caused by strong electromagnetic force, resulting in serious plastic deformation, which contributes to the hardness of the joints interface becoming stronger. Magnetic pulse welding resembles explosive welding, both of them are high energy rate welding. In the process of high energy rate forming, dislocation multiplication and grain refinement will occur at the interface. Due to the distinction of thermal expansion coefficient of Al-Mg dissimilar metal, obvious residual stress will also occur near the joint, which is the reason for the significant increase of interface hardness [23,33]. After heat treatment at 100 °C and 150 °C, the microhardness of the Al-Mg interface is slightly lower than the condition of as-welded, which is due to the effect of heat weakening the degree of interfacial work hardening and the release of the interface residual stress. After heat treatment at 200 °C, 250 °C and 300 °C, there is an apparent increasing of the microhardness near the interface than that as-welded, reaching more than 120 HV, which is contributed by the formation of Al and Mg intermetallic compounds at the interface that generated at this temperature.

#### 3.4.2. Tensile Shear Performance of Welded Joint

##### Tensile Shear Performance

Figure 11 is the physical drawing of tensile shear fracture of Al-Mg welded joints. For the specimens in the welded state, 100 °C and 150 °C, the joints did not crack in the tensile shear process, but the 1060 Al fractured, as shown in Figure 11a, which indicates that the strength of joints is higher. Tensile shear specimens cracked along the joints at 200 °C and 250 °C was shown in Figure 11b. It can be known from the microstructure analysis that intermetallic compounds are produced at the interface at 250 °C. With the increase of temperature, intermetallic compounds gradually increased. However, the intermetallic compounds are hard and brittle, leading to serious deterioration of mechanical properties of the joints. In the tensile shear test of specimens treated at 300 °C, cracks occurred in the joint part during the clamping process, indicating that the joint performance deteriorated seriously at this temperature. Figure 12a is the force-displacement curve of the welded parts in the tensile shear process, and Figure 12b is the maximum bearing capacity error diagram of specimens at different temperatures. In the as-welded, at 150 °C and 200 °C, the tensile shear properties of the joint are similar, the maximum tensile shear load is 4500 N, and the elongation is high, reaching 2.65 mm. At 200 °C and 250 °C, the sound of brittle fracture is heard during the tensile shear process, and the elongation is very low, which indicates that the intermetallic compounds have severe impact on the joints of tensile shear properties.

##### Fracture Analysis

Figure 13 shows the welded state and the tensile shear fracture morphology of 100 °C and 150 °C. The fracture of aluminum alloy is ductile fracture, and the fracture surface is dimple, with the increase of heat treatment temperature, the size of the dimple gradually decreases, which is a typical ductile fracture. The fracture morphology of Al-Mg tensile shear specimen along the joint position at 200 °C, 250 °C and 300 °C is shown in Figure 14. The fracture is a typical cleavage fracture, belonging to brittle fracture.

When the heat treatment temperature is 200 °C and 250 °C, it can be known that the interface composition of Al-Mg magnetic pulse welding joint is Mg-Al_12_Mg_17_-Al. The interface composition of Al-Mg magnetic pulse welding joint is Mg-Al_12_Mg_17_-Al_3_Mg_2_-Al, when heat treated at 300 °C. For the sake of studying the tensile shear fracture mechanism of Al-Mg joints at different heat treatment temperatures, EDS analysis was conducted on the typical positions of aluminum and magnesium sides of the fracture specimens, the results are shown in Table 4.

The main component of magnesium side fracture is Al_12_Mg_17_ and a small amount of Al at 200 °C and 250 °C, which is concluded from the results of EDS analysis. The aluminum side fracture is mainly made up of Al and contains slight Al_12_Mg_17_, suggesting that the interface of Mg-Al_12_Mg_17_-Al is fractured under tensile and shear loads. Al_12_Mg_17_ contains more magnesium and less aluminum, which results in a significant difference in thermal expansion coefficient between Al_12_Mg_17_ and Al, and a large residual stress at the interface. The interface between Al_12_Mg_17_ and Al is prone to cracking under applied load. EDS results of the 300 °C section show that the main component of magnesium side fracture is Al_12_Mg_17_ and a small amount of Al_3_Mg_2_, while the main component of aluminum side fracture is Al_3_Mg_2_ and a small amount of Al_12_Mg_17_. This shows that the Mg-Al_12_Mg_17_-Al_3_Mg_2_-Al interface is under the pull-shear load, the Al_12_Mg_17_-Al_3_Mg_2_ interface fractured. Figure 15 shows the tensile shear fracture mechanism of Al-Mg magnetic pulse welded joints under different temperatures. The bonding strength of the joints at 100 °C and 150 °C is higher than other states, and the fracture occurs along the aluminum base metal at 200℃ and 250 °C, only Al_12_Mg_17_ intermetallic compound exists at the interface. Under tensile shear loading, the joint cracks along the interface of Al_12_Mg_17_-Al. The interface microstructure of Mg-Al_12_Mg_17_-Al_3_Mg_2_-Al is formed at 300 °C, and the joints cracked along the interface of Al_12_Mg_17_ and Al_3_Mg_2_ under tensile shear loading.

## 4. Conclusions

In this paper, the dissimilar metal welding joints of Al-Mg MPW were prepared, and the joints were annealed at different temperatures. The microstructure evolution and mechanical properties of Al-Mg MPW joints were studied at different temperatures. The following conclusions can be drawn:(1)Al-Mg magnetic pulse welding joint is a wavy bond, there are ASBs on the Mg side of the joints, Al and Mg elements diffuse each other on the interface, and the diffusion layer thickens obviously after thermal insulation treatment. At 200 °C, Al_12_Mg_17_ begins to grow on the welding interface. After the magnetic pulse welding is completed, the hardness of joints interface increases obviously. As the heat treatment temperature rises, the hardness of joints interface decreases first and then increases.(2)Two intermetallic compounds, Al_12_Mg_17_ and Al_3_Mg_2_ are generated at the Al-Mg interface after holding at 300 °C. Al_12_Mg_17_ is generated at the Mg side first, and Al_3_Mg_2_ is generated at the Al side, forming the interface composition of Mg-Al_12_Mg_17_-Al_3_Mg_2_-Al.(3)In the tensile shear tests, Al-Mg magnetic pulse welded joints do not crack at 100 °C and 150 °C, but the 1060 Al base metal fracture occurs. At 200 °C and 250 °C, the joints fracture along the Al_12_Mg_17_-Al interface under tensile shear loads. After heat treatment at 300 °C, the joint cracks along the interface of Al_12_Mg_17_-Al_3_Mg_2_ intermetallic compounds under tensile and shear loads.

## Figures and Tables

**Figure 1 materials-15-05519-f001:**
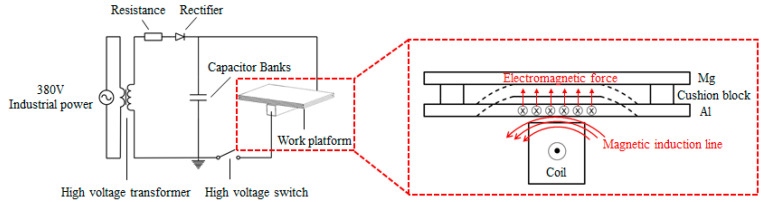
Schematic diagram of Al-Mg MPW.

**Figure 2 materials-15-05519-f002:**
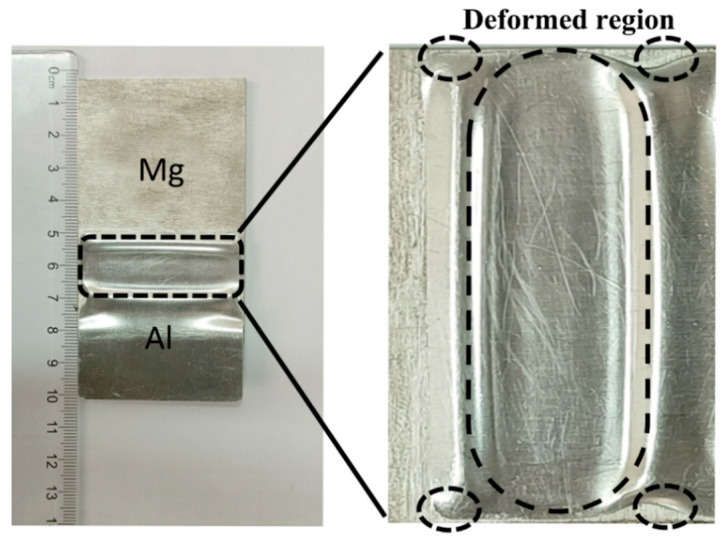
Macro morphology of Al-Mg MPW joint.

**Figure 3 materials-15-05519-f003:**
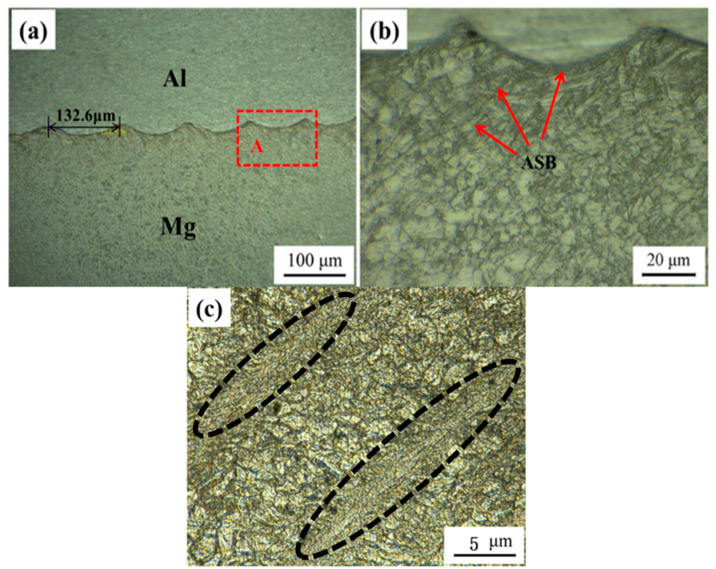
Optical micrographs of Al-Mg MPW joint (**a**) low magnification (**b**) high magnification on A area shown in (**a**). (**c**) Partial enlarged view of the gray part in (**b**).

**Figure 4 materials-15-05519-f004:**
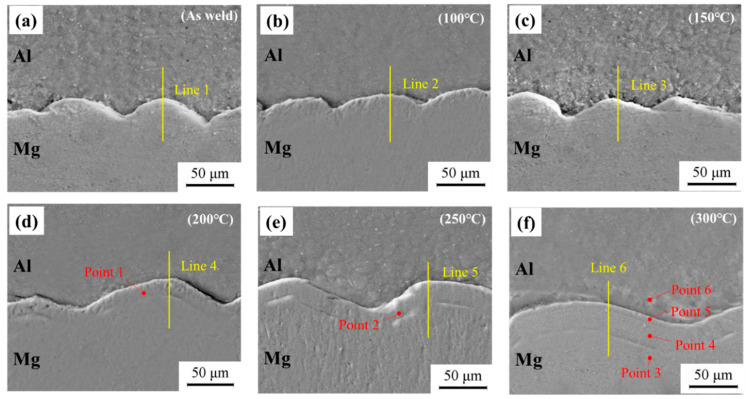
SEM images of Al-Mg interface after annealing at different temperatures (**a**) as-welded; (**b**) 100 °C; (**c**) 150 °C; (**d**) 200 °C; (**e**) 250 °C; (**f**) 300 °C.

**Figure 5 materials-15-05519-f005:**
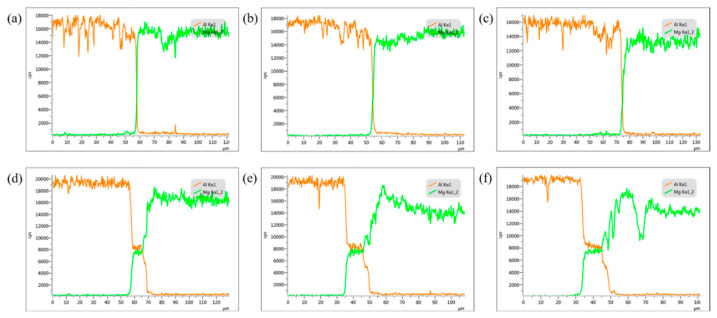
EDS line scan across the Al-Mg interface after annealing at different temperatures (**a**) As-welded; (**b**) 100 °C; (**c**) 150 °C; (**d**) 200 °C; (**e**) 250 °C; (**f**) 300 °C.

**Figure 6 materials-15-05519-f006:**
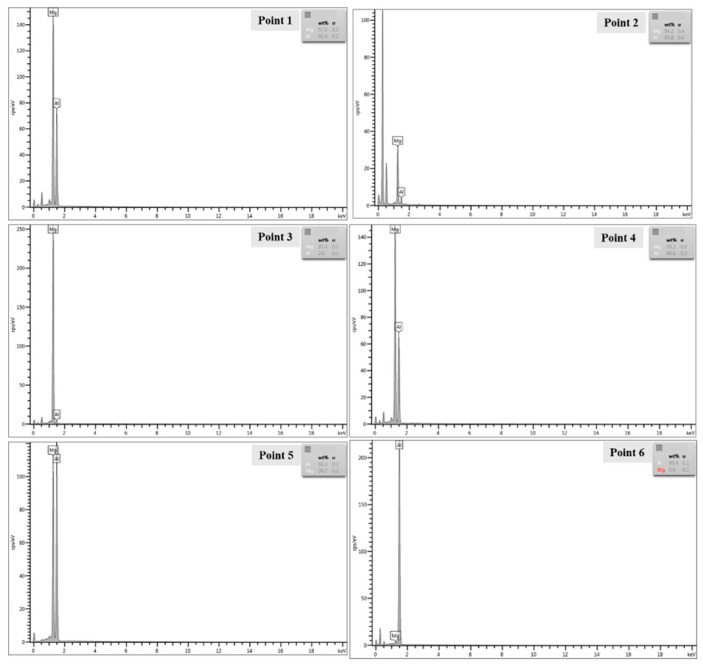
EDS spectra from the Al-Mg interface marked on Figure 4.

**Figure 7 materials-15-05519-f007:**
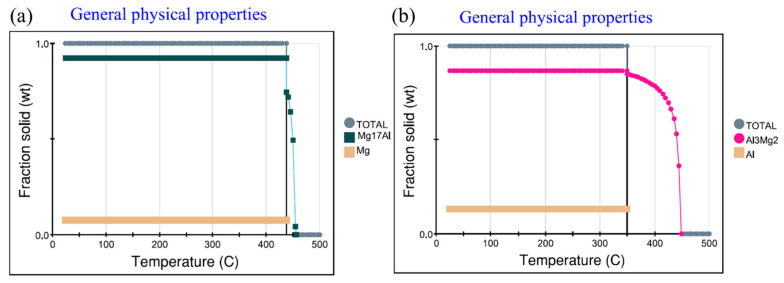
Non equilibrium solidification calculation of Al-Mg. (**a**) Point 4, calculation results of Mg rich area (**b**) Point 5, calculation results of Al rich area.

**Figure 8 materials-15-05519-f008:**
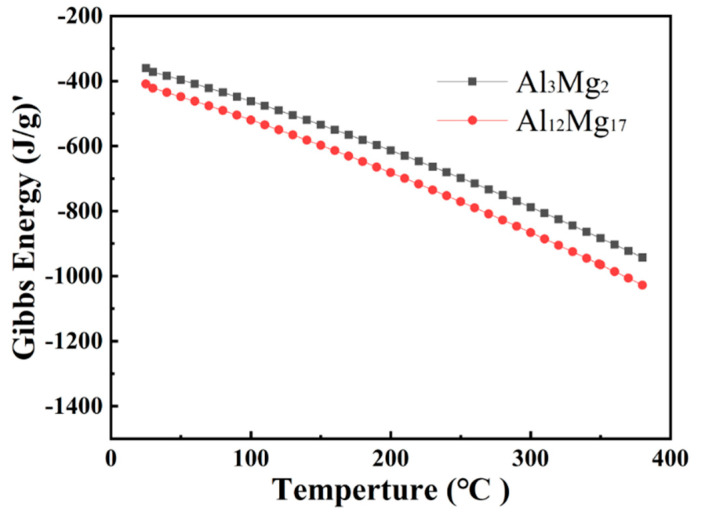
Gibbs free energy curves of Al_12_Mg_17_ and Al_3_Mg_2._

**Figure 9 materials-15-05519-f009:**
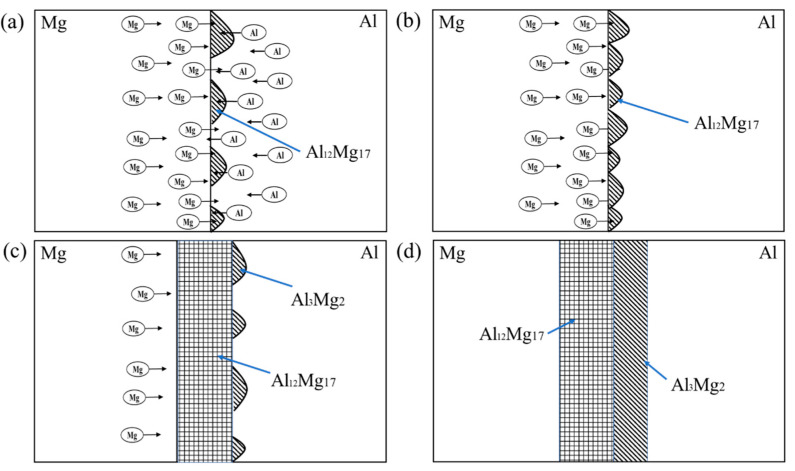
Growth model of the intermetallic compound at Al-Mg interface (300 °C). (**a**) As-welded, (**b**) 150 °C, (**c**) 200 °C, (**d**) 300 °C.

**Figure 10 materials-15-05519-f010:**
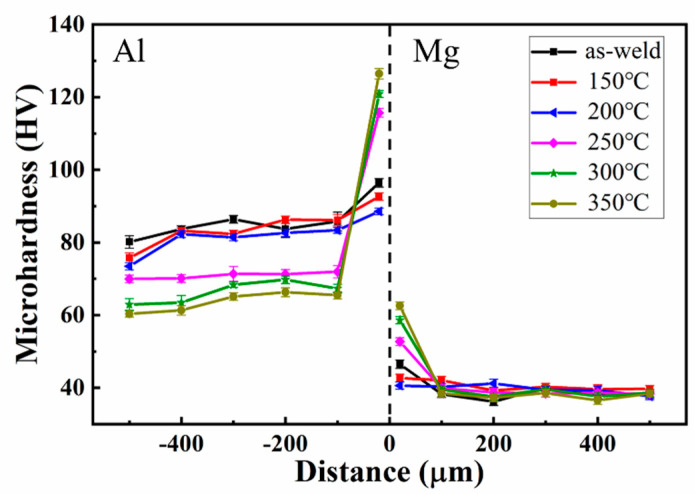
The microhardness profile across the Al-Mg interface after heat treatment at different temperatures.

**Figure 11 materials-15-05519-f011:**
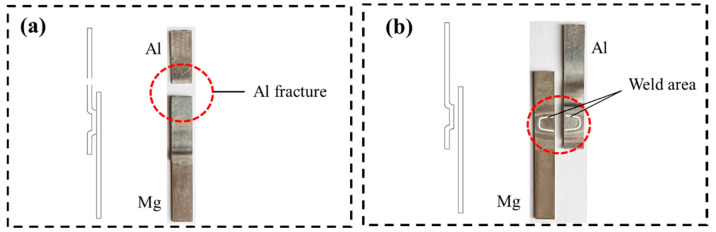
Tensile shear test fracture mode (**a**) As-welded, 100 °C and 150℃ (**b**) 150 °C, 250 °C and 300 °C.

**Figure 12 materials-15-05519-f012:**
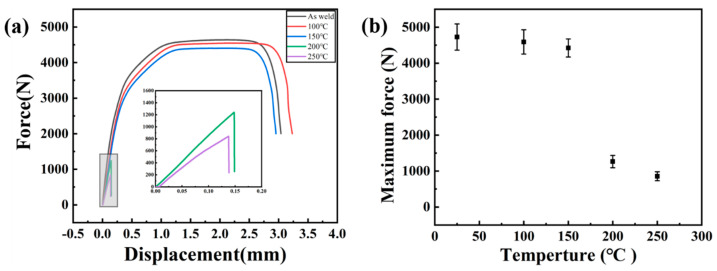
(**a**) The load–displacement curves of the Al-Mg joints after heat preservation treatment at different temperatures; (**b**) The effect of temperature on maximum shear force.

**Figure 13 materials-15-05519-f013:**
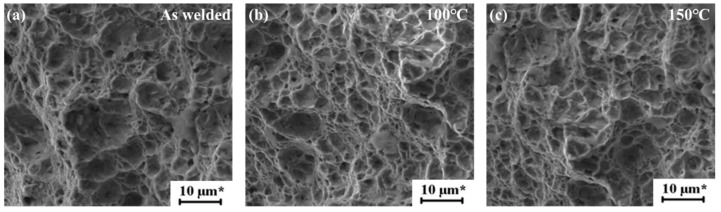
The fracture morphology of the 1060 Al after heat preservation treatment at different temperatures (**a**) As-welded; (**b**) 100 °C; (**c**) 150 °C.

**Figure 14 materials-15-05519-f014:**
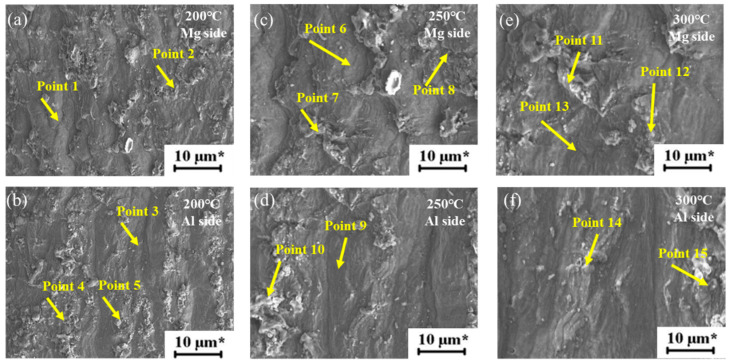
The fracture morphology of the Al-side and Mg-side of the Al/Mg welded joints after heat preservation treatment at different temperatures (**a**) Mg side, 200 °C; (**b**) Al side,200 °C; (**c**) Mg side, 250 °C; (**d**) Al side,250 °C; (**e**) Mg side, 300 °C; (**f**) Al side,300 °C.

**Figure 15 materials-15-05519-f015:**
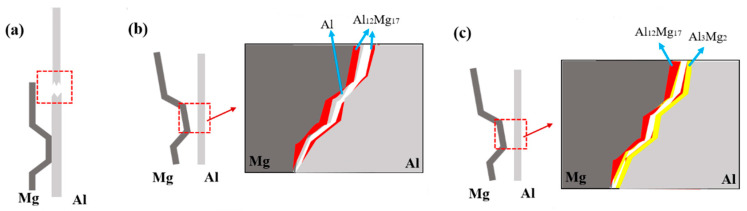
Schematic of Al/Mg welded joints tension shear fracture location: (**a**) As-welded, 100 °C and 150 °C; (**b**) 200 °C, 250 °C; (**c**) 300 °C.

**Table 1 materials-15-05519-t001:** Chemical composition (wt.%) of welding materials.

Material	Al	Mg	Si	Zn	Mn	Fe	Cu	Ti	Ca	Ni
1060 Al	99.6	0.03	0.25	0.05	0.03	0.35	0.05	0.03		
AZ31B Mg	2.5–3.5	Bal.	0.08	0.6–1.4	0.2–1.0	0.003	0.04		0.04	0.001

**Table 2 materials-15-05519-t002:** The point of EDS elements at the interface corresponds to Figure 4.

Point	Al (at.%)	Mg (at.%)	Material
Point 1	42.4	57.6	Al_12_Mg_17_
Point 2	45.8	54.2	Al_12_Mg_17_
Point 3	2.6	97.4	AZ31B
Point 4	40.8	59.2	Al_12_Mg_17_
Point 5	60.3	39.7	Al_3_Mg_2_
Point 6	99.4	0.6	1060Al

**Table 3 materials-15-05519-t003:** The diffusion factor and diffusion activation energy correspond to Al and Mg elements [32].

Elements	Diffusion Coefficient (m^2^/s)	Diffusion Activation Energy (J/mol)
Al	1.7 × 10^−4^	1.42 × 10^5^
Mg	1.5 × 10^−4^	1.35 × 10^5^

**Table 4 materials-15-05519-t004:** The point of EDS elements on the fracture surface of Al/Mg joints corresponds.

Point	1	2	3	4	5	6	7	8
Al (at.%)	100	41.4	100	40.9	41.8	100	42.2	40.7
Mg (at.%)	/	58.6	/	59.1	58.2	/	57.8	59.3
Point	9	10	11	12	13	14	15	
Al (at.%)	100	42.5	40.6	41.9	61.8	59.7	60.5	
Mg (at.%)	/	57.5	59.4	58.1	38.2	40.3	39.5	

## Data Availability

Not applicable.

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
