# Peer review of "Effect of Annealing Temperature on Microstructure and Properties of Al/Mg Magnetic Pulse Welding Joints"

_materials, 2022, doi:10.3390/ma15165519_

Round 1

Reviewer 1 Report

1. In the sentence in lines 37-38: "The application of these two alloys is limited [4]." is not clear to where the authors refer - do the authors mean the two alloys Al and Mg?

2. The kind of "jet" in line 118 ("high speed jet flow") should be specified. Is it ambient air jet or what?

3. The ASBs mentioned in line 140 should be further specified to what exactly correspond to the image in Figure 3(b): are the grey colored areas on the image?

4. A bit more detail explanation and clarification of the plots in Figure 7 should be added in the text in lines 216-218.

5. There is nothing mentioned in the text about Figure 8.

6. It is not clear what the authors mean by the term "dimple" in lines 302-304: "The fracture of the aluminium base material is a dimple,...". Please explain.

7. I think a better word for "insulated" (line 12, 342) would be "annealed".

8. The text needs syntax and grammar editing.

Author Response

Thank you for your letter and for the reviewers’ comments concerning our manuscript entitled “Effect of Temperature on Microstructure and Properties of Al/Mg Magnetic Pulse Welding Joints” (ID: metals- 1827899). Those comments are all valuable and very helpful for revising and improving our paper, as well as the important guiding significance to our researches. We have studied comments carefully and have made correction which we hope meet with approval. The main corrections in the paper and the responds to the reviewer’s comments are as flowing:

Responds to the reviewer’s comments:

1In the sentence in lines 37-38: "The application of these two alloys is limited [4]." is not clear to where the authors refer - do the authors mean the two alloys Al and Mg?

Lines 37-38 refer to the limitation of Mg/Al dissimilar metals welding technology. What we want to express is that the application of welded joints is limited due to the difficulty of welding Mg/Al dissimilar metals. We didn't express it clearly in the article. We have modified it in the article and cited relevant literature. Specific modifications are in lines 36-37 of the article.

2The kind of "jet" in line 118 ("high speed jet flow") should be specified. Is it ambient air jet or what?

The "jet" in line 118 is a metal jet, We have added specific explanations in the article. In the process of magnetic pulse welding, aluminum alloy plate and magnesium alloy plate collide violently, and the pressure at the impact is large, which will damage the metal surface. When the metal surface is damaged, metal debris will fall off, forming a metal jet. Specific modifications are in lines 121-124 of the article.

  1. The ASBs mentioned in line 140 should be further specified to what exactly correspond to the image in Figure 3(b): are the grey colored areas on the image?

Yes, the ASBs are the grey colored areas on the image, and we added a clearer picture to the article. Specific modifications are in lines 148-150 of the article.

  1. A bit more detail explanation and clarification of the plots in Figure 7 should be added in the text in lines 216-218.

We have explained the pictures in more detail in the article, Specific modifications are in lines 227-233 of the article.

  1. There is nothing mentioned in the text about Figure 8.

Due to our negligence, the introduction of picture 8 is missing. We have added this part to the article. Specific modifications are in lines 245-249 of the article.

  1. It is not clear what the authors mean by the term "dimple" in lines 302-304: "The fracture of the aluminium base material is a dimple,...". Please explain.

The fracture of aluminum alloy is ductile fracture because the fracture surface is dimple. It is not the fracture of aluminum base material is a dimple. The statement is wrong. We have made changes in the article. Specific modifications are in lines 320-321 of the article.

  1. I think a better word for "insulated" (line 12, 342) would be "annealed".

Yes, we made changes in the article. We have also made changes to other language issues in the article.

In other parts of the article, we have also made changes that are marked in the revised manuscript, which we have not specifically reflected here.

In addition to the above modifications, we have carefully checked and improved the English writing in the revised manuscript.

Thank you for your careful review. Your careful review has helped to make our study clearer and more comprehensive.

Reviewer 2 Report

The article presents a study on heat treatment of MPW dissimilar joints. The article is of academic importance but does not serve a practical purpose as the heat treatment weakens the joint. Anyhow, the observation is worth sharing with the readers. The article has a major issue with language. There are other issues as follows:

The language of the text is very poor at places. e.g the sentence in line 107-108, “ Al-Mg magnetic pulse welding specimen surface, the macro is shown in Figure 2, the  specimen surface quality is good, no collision crack and other defects, a certain degree of plastic deformation will happen in the welding area, have apparent necking and thinning.” Does not make any sense. The manuscript must be checked by a professional language editor.

The introduction part is very weak. It doesn’t give a full picture of the state-of-the-art. For a detailed description referhttps://doi.org/10.1016/j.jclepro.2015.03.042.

 The objective of the study and motivation are not clearly explained.  The heat treatment weakens the joint, so what is the purpose or outcome of this study?

 Provide a full form of every abbreviation when used for the first time.

The word ‘insulated’ seems out of context. Heat treated can be used in place of insulated.

Use as-welded in place of as-weld

The captions and text of all the figures should be of the same style and size. 

Author Response

Thank you for your letter and for the reviewers’ comments concerning our manuscript entitled “Effect of Temperature on Microstructure and Properties of Al/Mg Magnetic Pulse Welding Joints” (ID: metals- 1827899). Those comments are all valuable and very helpful for revising and improving our paper, as well as the important guiding significance to our researches. We have studied comments carefully and have made correction which we hope meet with approval. The main corrections in the paper and the responds to the reviewer’s comments are as flowing:

Responds to the reviewer’s comments:

1The language of the text is very poor at places. e.g the sentence in line 107-108, “ Al-Mg magnetic pulse welding specimen surface, the macro is shown in Figure 2, the  specimen surface quality is good, no collision crack and other defects, a certain degree of plastic deformation will happen in the welding area, have apparent necking and thinning.” Does not make any sense. The manuscript must be checked by a professional language editor.

According to your suggestion, we found relevant personnel to modify the language and polish the whole article. Specific modifications are in lines 110-114 of the article.

2The introduction part is very weak. It doesn’t give a full picture of the state-of-the-art. For a detailed description refer.

 https://doi.org/10.1016/j.jclepro.2015.03.042.

According to your suggestion, we quote this reference. In the article is references 8.

3The objective of the study and motivation are not clearly explained. The heat treatment weakens the joint, so what is the purpose or outcome of this study?

Annealing heat treatment is the main method for dissimilar metals after welding.

In this paper, the effect of annealing temperature on the microstructure and properties of Al/Mg magnetic pulse welded joints is studied, and the mechanism of element diffusion and compound precipitation at the Al/Mg interface at different temperatures is studied, which has a certain reference value for enriching the theory and application of Al/Mg magnetic pulse welding.

4Provide a full form of every abbreviation when used for the first time. The word ‘insulated’ seems out of context. Heat treated can be used in place of insulated. Use as-welded in place of as-weld

According to your suggestion, we changed each abbreviation to its full name when it first appeared. We replaced the original words with the words you suggested. There are many modifications in the article, we didn’t listed completely.

5The captions and text of all the figures should be of the same style and size.

We have changed the captions and text of all the figures in the article. But some pictures can't be changed, so we didn't modify them.

In other parts of the article, we have also made changes that are marked in the revised manuscript, which we have not specifically reflected here.

In addition to the above modifications, we have carefully checked and improved the English writing in the revised manuscript.

Thank you for your careful review. Your careful review has helped to make our study clearer and more comprehensive.

Reviewer 3 Report

The presented study entitled Effect of Temperature on Microstructure and Properties of Al/Mg Magnetic Pulse Welding Joints is interesting and logically organized. The abstract provides basic information about the solved problem and at the same time provides the basic results achieved by the authors. I have no comments on the experiment itself, the evaluation and justification of the results, and I recommend publishing the article in its current form.

Author Response

Thank you for your careful review. Your careful review has helped to make our study clearer and more comprehensive.

Reviewer 4 Report

The results are interesting and important for the investigators and technologists that work in the design and production of the Al/Mg welded joints. The authors used different experimental techniques to study the structural properties of Al/Mg welded joints and the impact temperature. The enhancement of the microstructure and mechanical properties of Al/Mg welded joints was not apparent under the temperature of 200. However, at 200 °C, and 300 °C, an intermetallic compound layer formed. The formation of an intermetallic compound is related to Al and Mg diffusion which is strongly dependent on the annealing temperature. These intermetallic compounds have an impact on the mechanical properties of the film.

In my opinion, this work fits well in the scope of Materials. Quality of presentation, discussion, and implications meet, overall, the standards of this Journal. However, there are several issues that the authors should address for the manuscript to be in a suitable form for publication. I provide a detailed list in attached

Author Response

Thank you for your letter and for the reviewers’ comments concerning our manuscript entitled “Effect of Temperature on Microstructure and Properties of Al/Mg Magnetic Pulse Welding Joints” (ID: metals- 1827899). Those comments are all valuable and very helpful for revising and improving our paper, as well as the important guiding significance to our researches. We have studied comments carefully and have made correction which we hope meet with approval. The main corrections in the paper and the responds to the reviewer’s comments are as flowing:

Responds to the reviewer’s comments:

1Is not clear how the authors from the EDS lines in table 2, identified the type of intermetallic compounds?

Due to our negligence in writing, EDS point scanning is written as line scanning. We identified the type of intermetallic compounds according to the composition ratio of atoms. For this part, we refer to an article: . The details are as follows:

Specific modifications are in lines 217 of the article.

Reference: https://doi.org/10.1016/j.jmrt.2019.05.017

2Fig.7 misses the description and should contain a clear analysis?

We have explained the pictures in more detail in the article, Specific modifications are in lines 227-233 of the article.

3In Fig.8 the authors should give a comparison between the evolution of the Gibbs energy as a function of the temperature of both compounds?

Due to our negligence, the introduction of picture 8 is missing. We have added this part to the article. Specific modifications are in lines 245-249 of the article.

4I recommend if it is possible to do x-ray diffraction on all the samples and based on the obtained results we will be sure about the type of intermetallic compounds.

We think your suggestion is correct. X-ray diffraction is a good method to determine the type of intermetallic compounds. Unfortunately, the samples have been lost for some reasons. According to your suggestion, we will improve the experimental method in the future experiments. Thank you very much for your suggestions, which are of great guiding significance to our future scientific research work.

5In Fig.10, the authors add a description, the previous version is not clear at all. As well as Fig.6 you don’t need all the range of the x-axis.

In picture 10, we added a description to distinguish between Al and Mg.

As for Figure 6, we know what you mean. The range of x-axis is too large, resulting in uneven image distribution. However, our samples have been lost, so it is not realistic to do another experiment. We accept your suggestions and pay more attention to these problems in our future work.

6In Fig.5 the authors should make it clear.

Figure 5 is unclear due to too many pictures. You can enlarge it in the word version.

7The introduction should be improved by one paragraph showing the importance of the interface to improve the mechanical properties and the interface stored high elastic strain. This elastic strain (residual stress) dependent on the type of the system such as: FCC/FCC (Cu/Co [1]), FCC/BCC (Cu/V [2]), (Cu/Nb [3]), BCC/BCC (Fe/W [4]), HCP/BCC (Nb/Zr [5-8]). The introduction should be enriched with strong references.

Good interface can improve mechanical properties. According to your suggestion, we have properly referred to relevant literature. In the article is references 15.

8The tensile test showed that the Al/Mg weld joints are resistant to fracture up to 200°C. The authors assume that this behavior is due to the formation of intermetallic compounds, but it can be also related to the transition of the interface from incoherent to semi-coherent [5, 8]. Because during annealing the redistribution of Al and Mg around the interface decreases the mismatch between the super-cell (hcp and fcc) [9].

Your suggestion is very important, but it takes a long time to do this experiment, and our samples have been lost. We add this part to the introduction of the paper and quote relevant literature. Your suggestion has important guiding significance for our future work. Specific modifications are in lines 62-65 of the article. In the article is references 16,17,18.

9The authors should describe all the figures and then provide the interpretation and output.

Due to our negligence, the introduction of picture 8 is missing. We have added this part to the article. In addition, we have also modified other pictures.

In other parts of the article, we have also made changes that are marked in the revised manuscript, which we have not specifically reflected here.

In addition to the above modifications, we have carefully checked and improved the English writing in the revised manuscript.

Thank you for your careful review. Your careful review has helped to make our study clearer and more comprehensive.

Round 2

Reviewer 4 Report

Dear editor 

Most of my concernes are adressed, so I suggest to accept the paper.

Please the reference n°17 is missing some details and it should be Acta Materialia volume 229, (2022), 117807. 

Best reagrds